# Genome-wide association identifies seven loci for pelvic organ prolapse in Iceland and the UK Biobank

Thorhildur Olafsdottir[1✉], Gudmar Thorleifsson[1], Patrick Sulem [1], Olafur A. Stefansson[1], Helga Medek[2], Karl Olafsson[2], Orri Ingthorsson[3], Valur Gudmundsson[3], Ingileif Jonsdottir [1,4,5], Gisli H. Halldorsson [1], Ragnar P. Kristjansson[1], Michael L. Frigge[1], Lilja Stefansdottir[1], Jon K. Sigurdsson[1], Asmundur Oddsson [1], Asgeir Sigurdsson[1], Hannes P. Eggertsson[1], Pall Melsted[1,6], Bjarni V. Halldorsson [1,7], Sigrun H. Lund [1], Unnur Styrkarsdottir[1], Valgerdur Steinthorsdottir [1], Julius Gudmundsson[1], Hilma Holm[1], Vinicius Tragante [1,8], Folkert W. Asselbergs [8,9,10], Unnur Thorsteinsdottir[1,4], Daniel F. Gudbjartsson [1,6], Kristin Jonsdottir[2], Thorunn Rafnar [1] & Kari Stefansson [1,4✉]

Pelvic organ prolapse (POP) is a downward descent of one or more of the pelvic organs, resulting in a protrusion of the vaginal wall and/or uterus. We performed a genome-wide association study of POP using data from Iceland and the UK Biobank, a total of 15,010 cases with hospital-based diagnosis code and 340,734 female controls, and found eight sequence variants at seven loci associating with POP ($P < 5 \times 10^{-8}$); seven common (minor allele frequency >5%) and one with minor allele frequency of 4.87%. Some of the variants associating with POP also associated with traits of similar pathophysiology. Of these, rs3820282, which may alter the estrogen-based regulation of *WNT4*, also associates with leiomyoma of uterus, gestational duration and endometriosis. Rs3791675 at *EFEMP1*, a gene involved in connective tissue homeostasis, also associates with hernias and carpal tunnel syndrome. Our results highlight the role of connective tissue metabolism and estrogen exposure in the etiology of POP.

[1] deCODE Genetics/Amgen, Sturlugata 8, 101 Reykjavik, Iceland. [2] Department of Obstetrics and Gynecology, Landspitali University Hospital, 101 Reykjavik, Iceland. [3] Department of Obstetrics and Gynecology, Akureyri Hospital, 600 Akureyri, Iceland. [4] Faculty of Medicine, School of Health Sciences, University of Iceland, 101 Reykjavik, Iceland. [5] Department of Immunology, Landspitali University Hospital, 101 Reykjavik, Iceland. [6] School of Engineering and Natural Sciences, University of Iceland, 101 Reykjavik, Iceland. [7] School of Science and Engineering, Reykjavik University, 101 Reykjavik, Iceland. [8] Department of Cardiology, Division Heart & Lungs, University Medical Center Utrecht, Utrecht University, Utrecht, The Netherlands. [9] Institute of Cardiovascular Science, Faculty of Population Health Sciences, University College London, London, UK. [10] Health Data Research UK and Institute of Health Informatics, University College London, London, UK. ✉email: thorhildur.olafsdottir@decode.is; kstefans@decode.is

Pelvic organ prolapse (POP) refers to the condition where one or more of the pelvic organs herniate to or beyond the vaginal opening and can involve the bladder, uterus, rectum, or post-hysterectomy vaginal cuff[1]. Symptoms affect quality of life and include bothersome sense of vaginal bulb, urinary or bowel symptoms, and sexual dysfunction[2–4]. The five stages of POP as defined by Pelvic Organ Prolapse Quantification system (POPQ)[5] range from stage 0; no prolapse, to stage IV which is complete eversion of the total length of the lower genital tract. Prevalence estimates for all stages of POP are 30–50% among postmenopausal women diagnosed on physical examination[6–8], but 7–26% if diagnosis is restricted to pelvic organs herniating beyond or at the hymenal remnant (stages II–IV)[9,10]. Symptom-based prevalence estimates are lower, or 3–6%[11,12] with incidence peaking between ages of 70 and 79[7]. Between 11 and 13% of women have had surgery for POP or related conditions by age 80 years[13–15]. POP is the leading indication for hysterectomy in postmenopausal women and accounts for 1 in 6 hysterectomies in all age-groups[16].

Risk factors for POP vary depending on POP classification. For POP stages II-IV (a descent at or beyond the hymenal remnant), risk factors are number of children delivered, vaginal delivery, advancing age and BMI[17], suggesting the role of tissue trauma, estrogen exposure and intra-abdominal pressure in the pathogenesis of POP. For all stages of POP, Hispanic ethnicity[6], lack of pelvic floor muscle strength[18] and family history[19] are also reported as risk factors, and previous hysterectomy is associated with severe POP[20]. For symptomatic POP, poor health status, constipation, or irritable bowel syndrome are also among reported risk factors[11].

Although the etiology of POP is not fully understood, and is likely multifactorial, abnormality in the connective tissue supporting the vagina and pelvic organs or in the muscles in the pelvic floor have been proposed to contribute to the pathophysiology of the condition[21,22]. The pelvic organs' support is dependent on interactions between the levator ani muscle and pelvic connective tissue, i.e. the ligaments holding the organs in alignment[23]. A higher prevalence and more recurrent POP is seen in women with joint hypermobility than in others[24,25]. Moreover, higher serum concentration of procollagen III[25] is found in women with joint hypermobility and in those with recurrent POP. It is, however, unclear whether such changes in collagen metabolism cause POP or are the result of a trauma[26,27].

Genetic factors have been estimated to explain 43% of the variation in risk of POP in a twin study[28]. However, previous candidate gene-, linkage- and genome-wide-association studies (GWASs) on POP, generally of small sample sizes, have not yielded sequence variants that associate with POP[26,29–31].

To provide insights into the etiology of POP, we performed a combined GWAS for POP in Iceland and the UK using data on 15,010 cases with hospital-based diagnosis code and 340,734 controls. We discovered eight variants at seven loci that associate with POP and point to a role of connective tissue metabolism and estrogen in the etiology of POP.

## Results

**Association analysis.** We performed a meta-analysis of two GWA-studies on POP, one from Iceland and the other from the UK (UK Biobank: UKB) with a combined sample of 15,010 cases and 340,734 controls of European ancestry (Supplementary Data 1). The Icelandic GWAS included 3409 cases and 131,444 controls and the corresponding numbers from UKB were 11,601 and 209,288. Cases were identified from hospital-based diagnosis records (International classification of disease (ICD) edition 10 code N81: Female genital prolapse) and controls were all females (see Methods for a detailed description of the Icelandic and UK datasets). To account for multiple testing, we used a weighted Bonferroni procedure based on sequence variant annotation[32] (Supplementary Data 2). A total of 245 variants at seven loci associate with POP at genome-wide significance (Fig. 1 and Supplementary Fig. 1, Supplementary Data 3). Conditional analysis at each locus identified a secondary signal at one (2p16.1) of the seven loci (Supplementary Data 4) resulting in eight distinct associations with POP (Table 1, Supplementary Fig. 2). All eight variants were nominally significant in both populations and accounting for multiple testing ($P$-value $> 0.05/8 = 6.25 \times 10^{-3}$), there is no significant heterogeneity in the effect estimates from the two datasets (Table 1). Seven out of the eight variants are

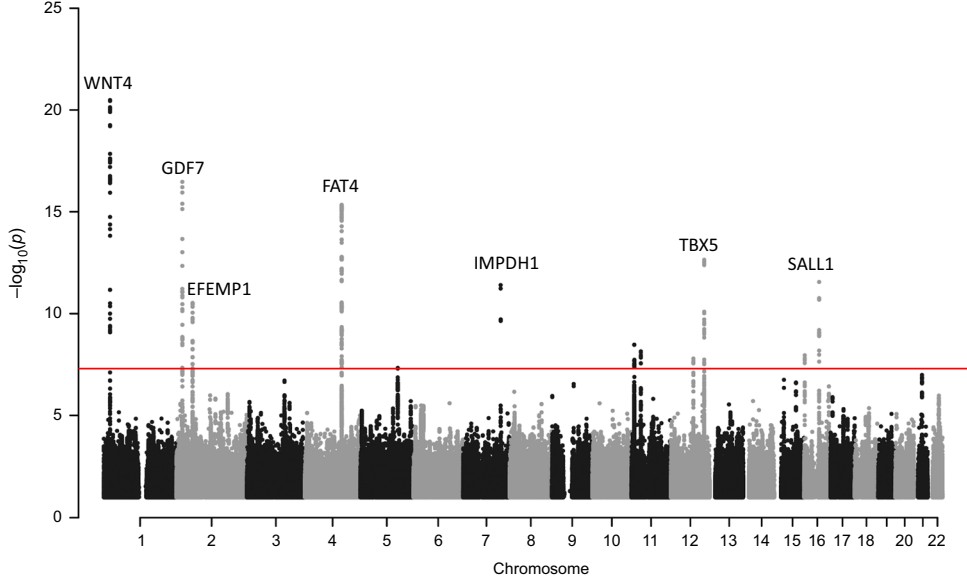

**Fig. 1 Manhattan plot of association results between sequence variants from Iceland and UK Biobank and pelvic organ prolapse in the meta-analysis.** The significance of association for each variant ($P$-values on -$\log_{10}$ scale) are plotted against the respective position on each chromosome. Red line indicates genome-wide significance level ($5 \times 10^{-8}$). Genes closest to the 7 loci of interest are annotated in the Figure. The plot was created using qqman: an R package for visualizing GWAS results using Q-Q and Manhattan plots[102]. Variants with imputation information >0.9 are displayed.

**Table 1 Association results for lead variants at loci reaching genome-wide significance in a meta-analysis of pelvic organ prolapse.**

| SNP | Pos hg38[a] and cytoband | Annotation | EAF (%) | EA | OA | Gene[b] | LD class | Pop. | OR[c] (95% CI) | P | Info | $P_{het}$ |
|---|---|---|---|---|---|---|---|---|---|---|---|---|
| rs3820282 | chr1:22,141,722 1p36.12 | Intron | 17.17 | T | C | WNT4 | 23 | Ice | 0.83 (0.78, 0.90) | $1.0 \times 10^{-6}$ | 1.00 | |
| | | | | | | | | UKB | 0.85 (0.82, 0.89) | $5.0 \times 10^{-16}$ | 1.00 | |
| | | | | | | | | Meta | 0.85 (0.82, 0.88) | $3.3 \times 10^{-21}$ | | 0.62 |
| rs9306894 | chr2:20,678,345 2p24.1 | 3′ UTR | 34.48 | G | A | GDF7 | 5 | Ice | 1.10 (1.04, 1.16) | $1.7 \times 10^{-3}$ | 1.00 | |
| | | | | | | | | UKB | 1.12 (1.09, 1.15) | $4.1 \times 10^{-15}$ | 1.00 | |
| | | | | | | | | Meta | 1.11 (1.09, 1.14) | $3.4 \times 10^{-17}$ | | 0.51 |
| rs1430191 | chr2:55,806,292 2p16.1 | Intergenic | 47.85 | T | C | EFEMP1 | 3 | Ice | 1.15 (1.08, 1.22) | $5.4 \times 10^{-6}$ | 1.00 | |
| | | | | | | | | UKB | 1.07 (1.04, 1.10) | $5.2 \times 10^{-6}$ | 0.99 | |
| | | | | | | | | Meta | 1.09 (1.06, 1.12) | $1.0 \times 10^{-9}$ | | 0.034 |
| rs3791675 | chr2:55,884,174 2p16.1 | Intron | 21.07 | T | C | EFEMP1 | 30 | Ice | 0.81 (0.74, 0.87) | $9.3 \times 10^{-8}$ | 1.00 | |
| | | | | | | | | UKB | 0.88 (0.85, 0.92) | $6.0 \times 10^{-12}$ | 1.00 | |
| | | | | | | | | Meta | 0.87 (0.84, 0.90) | $2.7 \times 10^{-17}$ | | 0.034 |
| rs7682992 | chr4:126,037,217 4q28.1 | Intergenic | 20.63 | T | A | FAT4 | 78 | Ice | 1.21 (1.13, 1.29) | $1.0 \times 10^{-8}$ | 1.00 | |
| | | | | | | | | UKB | 1.11 (1.07, 1.15) | $6.7 \times 10^{-10}$ | 1.00 | |
| | | | | | | | | Meta | 1.13 (1.10, 1.16) | $4.5 \times 10^{-16}$ | | 0.027 |
| rs72624976 | chr7:128,392,779 7q32.1 | 3′ UTR | 4.87 | T | C | IMPDH1 | 5 | Ice | 0.74 (0.65, 0.84) | $3.4 \times 10^{-6}$ | 1.00 | |
| | | | | | | | | UKB | 0.82 (0.76, 0.88) | $1.0 \times 10^{-7}$ | 0.94 | |
| | | | | | | | | Meta | 0.80 (0.75, 0.85) | $4.0 \times 10^{-12}$ | | 0.18 |
| rs1247943 | chr12:114,235,616 12q24.21 | Intergenic | 46.03 | G | A | TBX5 | 10 | Ice | 1.10 (1.05, 1.17) | $3.3 \times 10^{-4}$ | 1.00 | |
| | | | | | | | | UKB | 1.09 (1.06, 1.12) | $1.5 \times 10^{-10}$ | 1.00 | |
| | | | | | | | | Meta | 1.09 (1.07, 1.12) | $2.3 \times 10^{-13}$ | | 0.73 |
| rs12325192 | chr16:51,454,218 16q12.1 | Intergenic | 17.90 | T | C | SALL1 | 20 | Ice | 0.93 (0.86, 1.00) | 0.045 | 1.00 | |
| | | | | | | | | UKB | 0.89 (0.86, 0.92) | $1.3 \times 10^{-11}$ | 1.00 | |
| | | | | | | | | Meta | 0.89 (0.87, 0.92) | $2.8 \times 10^{-12}$ | | 0.32 |

P P-value, OR odds ratio, Phet P-value for heterogeneity in the effect estimate between the Icelandic and UK Biobank data, EAF effect allele frequency (average of ICE and UKB), EA effect allele, OA other allele, info imputation information score, LD linkage disequilibrium, Pop. population. Results of the meta-analysis of the GWA-studies from Iceland (Ice) and UKB are shown in italics.
[a]Variant positions (chromosome:position) are according to GRCh38/hg38.
[b]For intergenic variants, nearest gene is reported.
[c]Odds-ratios correspond to effect alleles.
[d]Two distinct signals were detected at the 2p16.1 locus, rs3791675 and rs1430191. OR and P for rs3791675 are conditioned on rs1430191 and vice versa.

common, with minor allele frequency (MAF) range of 17–48%, and one is of low frequency (MAF = 4.87%). The number of variants that correlate with the lead variants ($r^2 > 0.8$) at these loci ranges from 3 to 78 (Table 1, Supplementary Fig. 2). No previously suggested POP variants from candidate gene, linkage or GWAS studies[26,29–31] associate with POP (Supplementary Data 5, 6).

We assessed the robustness of the POP association results for the eight lead variants in a group of more stringently diagnosed POP cases that are defined by procedure codes specific to the treatment of POP in addition to the ICD10 code N81 (Supplementary Data 7). The statistical power for this analysis comes mainly from the UKB data because information on POP related surgeries was available for 61% of the ICD-coded POP cases in UK, but only 13% of the Icelandic POP cases. The effect sizes tended to be greater for the association using cases that had undergone surgery compared to those diagnosed only based on ICD code (Supplementary Data 8). This most likely reflects increased severity of POP among those that had undergone surgery or a greater number of falsely diagnosed POP cases among the ICD based cases. We conclude that our results based on ICD codes alone are robust as the effects are in the same direction and the effect sizes are not substantially different from those using surgically treated POP cases.

We used LD score regression to estimate the SNP heritability of POP[33]. Using LD scores for about 1.1 million variants found in European populations we estimated SNP heritability in the meta-analysis to be 12.4% (95% CI 9.9–14.8%).

**Functional annotation and biological inference of risk loci.** None of the lead POP variants are coding or in high LD ($r^2 > 0.8$) with coding variants (Supplementary Data 3) and none was highly correlated with the top cis-eQTL for neighboring genes in any of the available tissues using mRNA expression data from the GTEx database and our own RNA-sequence data on Icelandic samples from blood (13,162 individuals) and adipose tissue (749 individuals).

To gain a further understanding of the variants associating with POP we explored how they correlate with other traits in combined data from Iceland and UKB – a database with approximately 800 traits (P-value threshold = $0.05/800 = 6.3 \times 10^{-5}$) – in addition to looking up all variants and their correlates ($r^2 > 0.5$) in the GWAS-catalog[34] (Supplementary Data 9, 10). As we observed overlaps of genetic association to different traits we conducted adjusted association analyses (co-localization analyses) at the POP loci to identify association results that are consistent with a single signal representation (see Methods).

Six of the eight POP variants also associate with a total of 13 other phenotypes. Three of those variants associate with six phenotypes with pathogenesis that may be the same as that of POP. Two POP variants associate with conditions related to estrogen exposure, rs3820282 at the *WNT4* locus (leiomyoma of uterus, gestational duration and endometriosis) and rs12325192 at the *SALL1* locus (leiomyoma). The third, rs3791675 at the *EFEMP1* locus, associates with conditions related to aberrant connective tissue function (hernias and carpal tunnel syndrome) (Table 2).

Secondary, non-symptomatic diagnosis of POP through conditions such as leiomyoma and endometriosis may introduce a confounding of or biases in the association results from our GWAS meta-analysis. We therefore tested the association of our eight POP variants with POP after excluding endometriosis and leiomyoma cases from the analysis. This did not affect effect or significance substantially (Supplementary Data 11) indicating that the POP associations are independent of endometriosis and leiomyoma.

**POP associated loci.** The most significant POP association is with rs3820282–T ($P = 3.3 \times 10^{-21}$, OR = 0.85), located in intron 1 of *WNT4*, which encodes a protein involved in development of the female reproductive tract[35]. Loss of *WNT4* function in humans leads to underdevelopment and sometimes absence of the uterus and vagina[36]. Variants correlated ($r^2 > 0.8$) with the POP-protecting allele rs3820282–T have been associated with increased risk of endometriosis[37], leiomyoma[38], increased gestational duration[39] and with decreased bone mineral density[40] (Supplementary Data 10), all of which we replicate (Supplementary Data 9). Rs3820282–T also associates with lower number of children in our combined data (Supplementary Data 9). Our co-localization analyses revealed that apart from the bone mineral density and number of children signals, the association results for leiomyoma, gestational duration, and endometriosis are likely to be the same signal as the POP association at the locus (Supplementary Data 12 and Table 2).

Rs12325192 is located near *SALL1* and the POP-protecting allele rs12325192–T is correlated ($r^2 = 0.94$) with rs66998222, a variant we have previously reported to associate with leiomyoma[38]; a trait that is positively correlated with estrogen exposure (Table 2). Mutations in *SALL1* have been associated with Townes-Brocks syndrome, a condition associated with renal malformation, suggesting a role of *SALL1* in genitourinary development[41].

Two distinct POP variants, rs3791675 and rs1430191, are located in and near *EFEMP1* (also known as *FBLN3*), a gene encoding fibulin-3. Fibulins are components of microfibrils, the building blocks of elastic fibers that are generated in fibroblasts[42] and provide tissue elasticity. *EFEMP1* is expressed in mouse connective tissue and network analyses suggest a role for *EFEMP1* in connective tissue maintenance/homoeostasis[43]. Rare mutations in *EFEMP1* have been found to cause autosomal dominant Doyne honeycomb degeneration of retina, characterized by yellow–white deposits known as drusen that accumulate beneath the retinal pigment epithelium[44]. The strongest POP association at the locus is with rs3791675–T ($P = 2.7 \times 10^{-17}$, OR = 0.87). Its correlates ($r^2 > 0.8$) have been associated with various traits in Europeans, including less height[45,46], smaller BMI-adjusted waist circumference[47,48] and protection against inguinal hernia[43] (Supplementary Data 10). We replicate these associations with our data (Table 2). In addition, rs3791675–T associates with protection against various hernias; inguinal hernia, femoral hernia, umbilical, and ventral hernia, consistent with the hypothesis of a similar proposed collagen pathophysiology for POP and hernias[49–52]. Rs3791675–T also associates with protection against diverticular disease, a disease in which abnormal collagen and decreased tensile strength of the colonic wall have been proposed to contribute to pathogenesis[53]. We furthermore found rs3791675–T to associate with increased risk of carpal tunnel syndrome – a condition also linked to connective tissue metabolism[54] – higher pulse pressure, and the lung function ratio forced expiratory volume (FEV1)/forced vital capacity (FVC) (Supplementary Data 9). According to our co-localization analyses, association results for inguinal hernia, height, FEV1/FVC, carpal tunnel syndrome, ventral hernia, and waist circumference (Supplementary Data 12 and Table 2), are consistent with a single signal origin.

Rs9306894, located in the 3′UTR of *GDF7* (also known as *BMP12*), associates with POP. *GDF7* encodes a secreted ligand of the TGF–beta superfamily of proteins and is thought to be involved in tendon and ligament formation and repair[55,56]. The

**Table 2 A summary of previously reported (*) and novel associations of POP variants with other traits.**

| SNP [EA/OA] | Position hg38 | EAF (%) | Gene | Trait | OR/$\beta$ | (95% CI) | P | $r^2$ | Published*/Top SNP | PubmedID |
|---|---|---|---|---|---|---|---|---|---|---|
| rs3820282 [T/C] | 1:22141722 | 17.17 | WNT4 | POP | 0.85 | (0.82, 0.88) | $3.3 \times 10^{-21}$ | 0.877 | rs10917151* | 30194396 |
| | | | | Leiomyoma | 1.12 | (1.09, 1.16) | $5.3 \times 10^{-13}$ | 0.878 | rs56318008* | 28877031 |
| | | | | Gestational duration | 0.058 | (0.04, 0.08) | $3.9 \times 10^{-11}$ | 0.945 | rs12037376* | 28537267[a] |
| | | | | Endometriosis | 1.12 | (1.06, 1.18) | $1.1 \times 10^{-5}$ | | | |
| rs9306894 [G/A] | 2:20678345 | 34.48 | GDF7 | POP | 1.11 | (1.09, 1.14) | $3.4 \times 10^{-17}$ | 0.996 | rs9306895* | 29892016[b] |
| | | | | Prostate cancer | 1.07 | (1.04, 1.11) | $6.2 \times 10^{-6}$ | 0.620 | rs7255* | 27527254 |
| | | | | BE and EA | 1.09 | (1.05, 1.14) | $4.2 \times 10^{-5}$ | | | |
| rs3791675 [T/C] | 2:55884174 | 21.07 | EFEMP1 | POP | 0.87 | (0.88, 0.93) | $2.7 \times 10^{-17}$ | | | |
| | | | | Inguinal hernia[c] | 0.82 | (0.80, 0.84) | $6.2 \times 10^{-44}$ | 0.960 | rs59985551* | 26686553 |
| | | | | Height | −0.081 | (−0.07, −0.10) | $6.7 \times 10^{-27}$ | 0.950 | rs3791679* | 25282103 |
| | | | | FEV1/FVC | 0.033 | (0.03, 0.04) | $4.1 \times 10^{-24}$ | 1 | rs3791675 | |
| | | | | Carpal tunnel s. | 1.11 | (1.08, 1.15) | $9.0 \times 10^{-11}$ | 0.960 | rs191535629 | |
| | | | | Ventral hernia | 0.86 | (0.81, 0.91) | $5.9 \times 10^{-8}$ | 0.950 | rs3791679 | |
| | | | | Waist circumference | −0.036 | (−0.04, −0.03) | $1.8 \times 10^{-35}$ | 0.950 | rs3791679* | 25673412 |
| rs7682992 [T/A] | 4:126037217 | 20.63 | FAT4 | POP | 1.13 | (1.10, 1.16) | $4.6 \times 10^{-16}$ | | | |
| | | | | Stress incontinence[d] | 1.11 | (1.06, 1.15) | $6.4 \times 10^{-7}$ | 0.640 | rs2893158 | |
| rs1247943 [G/A] | 12:114235616 | 46.03 | TBX5 | POP | 1.09 | (1.07, 1.12) | $2.3 \times 10^{-13}$ | | | |
| | | | | BPH | 0.92 | (0.90, 0.94) | $4.2 \times 10^{-12}$ | 0.990 | rs2555019* | 30410027[e] |
| | | | | Prostate cancer | 1.05 | (1.02, 1.08) | $1.8 \times 10^{-3}$ | 0.814 | rs1270884* | 29892016[f] |
| rs12325192 [T/C] | 16:51454218 | 17.90 | SALL1 | POP | 0.89 | (0.87, 0.92) | $2.8 \times 10^{-12}$ | 0.936 | rs66998222* | 30194396 |
| | | | | Leiomyoma | 0.91 | (0.88, 0.94) | $4.0 \times 10^{-9}$ | | | |

P P-value for fixed effects meta-analysis, OR odds ratio, $\beta$ effect estimate (italic), EAF effect allele frequency (average of ICE and UK data), $r^2$ linkage disequilibrium between POP index variant and the corresponding variant listed, BE and EA Barrett's oesophagus and esophageal adenocarcinoma combined, BPH benign prostatic hyperplasia, FEV1/FVC the ratio of forced expiratory volume and forced vital capacity.

Shown are association results consistent with a single signal origin, after performing conditional analysis using data from ICE and UKB (height and gestational duration is Iceland only, BE and EA is UKB only). Top variants are defined for each trait within ±500 kb from the POP variant at each locus. Variant locations are according to GRCh38/hg38. P and OR for rs3791675 are adjusted for the secondary signal rs1430191 in the POP phenotype (see Supplementary Data 4).

[a]See also Rahmioglu et al.[67].

[b]Originally reported in Amin Al Olama et al.[103].

[c]rs59985551 is top variant for inguinal hernia in the combined data from ICE and UKB. The previously reported variant at the locus, rs2009262, associates with POP with $P = 1.63 \times 10^{-36}$ (OR = 0.83) and is in $r^2 = 0.85$ with rs3791675.

[d]319 of the cases diagnosed with stress incontinence (ICD10 N393) are also diagnosed with POP.

[e]Two independent signals were reported at the 12q24.21 locus, rs2555019 and rs8853. Only rs2555019 is correlated with the POP variant rs1247943 ($r^2 = 0.002$ for rs8853).

[f]Originally reported in Eeles et al.[60].

protein has been found to play a crucial role in tenogenesis of mesenchymal stem cells and is used in tissue-engineering to treat tendon injury[57]. A correlated variant near *GDF7* (rs2289081–C, $r^2 = 0.85$) has been associated with decreased pulse pressure[58] and rs7255–T ($r^2 = 0.62$ to rs9306894) has been associated with increased risk of Barrett's oesophagus and esophageal adenocarcinoma combined (BE and EA) (Supplementary Data 10), which we replicate (Supplementary Data 9) but co-localization analysis only supports a single signal origin for BE and AE combined (Supplementary Data 12). We replicate the previous association of a highly correlated variant rs9306895 ($r^2 = 0.996$) with prostate cancer[59] ($P = 6.15 \times 10^{-6}$, OR = 1.07) (Table 2).

For the POP variant rs1247943, close to *TBX5* (distance: 172 Kb), we replicate the previous associations of two highly correlated variants with the POP risk-increasing allele; rs1270884 ($r^2 = 0.81$) and rs2555019 ($r^2 = 0.99$) with increased risk of prostate cancer[60] and lowered risk of benign prostatic hyperplasia[61] (Table 2).

Rs7682992–T, the POP variant close to the *FAT4* gene (distance: 0.5 Mb), associates with increased risk of stress incontinence (Table 2). *FAT4* is part of the planar cell polarity pathway that controls tissue organization; loss of *FAT4* in mice leads to cystic kidney disease[62].

Rs72624976 is located in the 3′UTR of *IMPDH1*, one of two target genes for the immunosuppressant drug mycophenolic acid[63]. Mutations in *IMPDH1* have been associated with the eye disorders retinitis pigmentosa[64] and Leber´s congenital amaurosis[65].

**Comorbidity**. To test further for comorbidity between POP and other traits, we collected sequence variants reported to associate with traits that associate with one or more of our POP variants: leiomyoma (N = 32)[38,66], gestational duration (N = 5)[39], endometriosis (N = 27)[67], BE and AE combined (N = 14)[68], prostate cancer (N = 145)[59], inguinal hernia (N = 4)[43], height (N = 3,290)[69], FEV1/FVC (N = 52)[70], BMI-adjusted waist circumference (N = 70)[48] and benign prostatic hyperplasia (N = 23)[61]. Of the 3,652 unique variants collected, 17 were correlated within height ($r^2 > 0.8$). We tested these 3,635 (3652–17) independent sequence variants for association with POP. As 44 variants were correlated across traits ($r^2 > 0.8$), we used the significance threshold of $P < 1.4 \times 10^{-5}$ (0.05/3591) (3652-17-44 = 3591) (Supplementary Data 13–22). We found variants at nine loci to associate with POP; five of which associate at genome-wide significance with POP, three that were previously associated with height ($P < 1.4 \times 10^{-5}$) at *TXNDC5* (6p24.3), *SLC12A2* (5q23.3) and *LOXL1* (15q24.1) (Supplementary Data 19) and one at *WT1* that associates with inguinal hernia (11p13) (Supplementary Data 18).

We further used the variants collected for each of the ten traits as instruments to explore a possible causal relationship between each trait and POP[71]. By regressing the effect estimates of each group of variants (trait-specific) on POP on the effect estimates of those variants on the corresponding traits, using AF × (1-AF) as a weight[72], we did not observe a correlation between the effect estimates (Supplementary Fig. 3).

We also plotted the POP effects of the eight POP variants against their effects on the 13 traits listed in Table 2 (Iceland and UKB combined, apart from height and gestational duration which are data from Iceland) (Supplementary Fig. 4). It is apparent that although a handful of variants associate with POP and other phenotypes, there is little overlap between the variants associating with POP and the other phenotypes and little correlation between effect estimates. We are not aware of previous literature findings showing associations of these phenotypes within POP cases and their family members.

**BMI is not associated with POP at the genetic level**. We screened for evidence of a genetic relationship of POP and two of the most consistently reported risk factors for POP in addition to age; BMI and number of children. We verified that variants reported to associate with BMI[73,74] or number of children[75] do not associate with POP (Supplementary Data 23, 24) and that adding BMI or number of children separately as covariates in our models did not affect the effect sizes or significance of the eight lead variants when tested for association with POP (Supplementary Data 25). Furthermore, no correlation was found between effect estimates of BMI- or number of children sequence variants and their effects on POP and vice versa (Supplementary Figs. 5, 6). Using polygenic risk scores for POP, we saw little evidence of association with number of children (P-value = 9.5 × 10$^{-5}$, beta = 0.015) in Iceland only and none with BMI (Supplementary Data 26). For BMI, we found no evidence of causal relationship with POP at the genetic level. With that in mind, and since only one of the POP variants (rs3820282) associates with number of children we do not find strong support for a causal pathway or a common third factor affecting POP and those two traits.

## Discussion

We conducted a meta-analysis of two GWAS for POP and discovered eight variants at seven loci that associate with POP. Two sequence variants, rs3820282 at *WNT4* and rs12325192 near *SALL1* also associate with other traits that are strongly affected by estrogen exposure, i.e. leiomyoma (rs3820282 and rs12325192), gestational duration (rs3820282) and endometriosis (rs3820282). Of the 23 correlated variants at the 1p36.12 locus, rs3820282 has the best functional candidacy. Rs3820282–T resides in a region marked by mono-methylation of histone H3 at lysine residue K4 (H3K4me1), a characteristic of enhancers, and is predicted to alter a conserved binding site for the transcription factors (TFs) ESR1 and ESR2[76] supported by ChIP-seq data in breast and bone cell lines[77]. This suggests that the variant may alter the estrogen-based regulation of *WNT4* or adjacent genes[39]. A previous report showed chromatin looping between the region containing rs12038474 (LD to rs3820282; $r^2 = 0.81$) and *CDC42* in endometrial adenocarcinoma[78]. Our analysis of available Hi-C data[79] along with enhancer-gene predictions[80] suggests *WNT4*, *HSPG2*, and *CDC42* as possible enhancer-gene targets, using relevant cell-types and tissues. Based on the role of *WNT4* in female sex organ development[35], *WNT4* can be considered to be a strong candidate gene at the locus. One of the eight POP variants, rs3791675 at *EFEMP1*, associates with traits with proposed collagen pathophysiology, i.e. inguinal hernia, ventral hernia and carpal tunnel syndrome. Furthermore, two genes at POP associated loci, *GDF7* and *EFEMP1*, have functions related to connective tissue metabolism.

We find that the overlap between POP variants and other traits are limited to distinct signals and thus do not allow for strong inference regarding common pathways or causal relationships. However, the pleiotropy found at three of the POP loci, where the associating traits have similar proposed pathophysiology as POP, together with the known functions of the proposed target genes, may fuel further explorations of the possible role of estrogen exposure and connective tissue metabolism in the etiology of POP. Through analysis of variants that are reported to associate with conditions that associate with one or more of our POP variants, we identified four additional POP variants, three that associate with height and one with inguinal hernia.

In mice, a POP phenotype has been associated with knock-down of fibulin 3 (*EFEMP1*), lysyl oxidase like 1 (*LOXL1*), fibulin 5 (*FBLN5*), and homeobox A11 (*HOXA11*) as summarized in a

recent review on key genes and pathways for POP[81]. Of those four genes, variants at the *EFEMP1* locus associate with POP in our data and, through analysis of height variants, also a variant at the *LOXL1* locus; rs12440667–T ($P = 3.6 \times 10^{-6}$, OR = 0.94) (Supplementary Data 19, Supplementary Fig. 7). *EFEMP1* encodes fibulin–3 but members of the fibulin family of extra-cellular matrix (ECM) proteins have been found to be important in elastic fiber assembly[82,83]. The two *EFEMP1* mouse knockout studies reported to date[84,85] demonstrate an abnormality in the integrity of elastic fibers in fascia connective tissue and connective tissue of the vaginal wall with abnormal pelvic organ support in addition to increased activity of matrix metalloprotease (MMPs) that degrade matrix collagen in connective tissue. These mice also displayed an early onset of aging-associated traits including decreased body mass, lordokyphosis, reduced hair growth, and generalized fat, muscle and organ atrophy, as well as reduced lifespan. The POP variants in and near to *EFEMP1* did not associate with traits likely to cause lordokyphosis (any fracture and osteoporosis in the ICE-UKB data, and BMD (hip and spine measures)), BMI (both sexes and females only) or lifespan (females) in the Icelandic data (Supplementary Data 27).

In this paper, we report the first set of genome-wide significant POP variants identified through GWAS. The genetic overlap observed between POP and several traits with similar patho-physiology point towards a role of estrogen exposure and con-nective tissue metabolism in the etiology of POP. The results provide new insights for future research to further the under-standing of POP.

## Methods

**Datasets.** The meta-analysis combined the results of two GWA studies of pelvic organ prolapse (POP). The Icelandic data originates from Landspitali – The National University Hospital Inpatient Registry from January 1983 to August 2018. A total of 3,699 women had a POP diagnosis (ICD10 code N81 or ICD9 code 618). Controls were recruited through different genetic research projects at deCODE genetics. We had genotype information for 92% of the POP cases, so that the Icelandic data consisted of 3409 cases and 131,444 female controls. The UK Biobank data (UKB) consists of 11,601 cases and 209,228 controls of European ancestry (self-reported white British with similar genetic ancestry based on prin-cipal component analysis and with consistent reported and genetically determined gender)[86], recruited between 2006 and 2010 aged 40–69, and with follow-up until 2016[87]. The UKB cases had ICD10 code N81 in hospital inpatient records, with data dating back to between 1981 and 1997, depending on the external source (Hospital Episode Statistics from England (89% of participants), Patient Episode Database for Wales (7%) and Scottish Morbidity Record (7%)) and include primary and secondary diagnosis of POP (http://biobank.ctsu.ox.ac.uk/showcase/docs/inpatient_mapping.pdf).

Hospital-based POP diagnosis are likely to be stage II or greater, according to the POPQ standardized grading system[5]. Furthermore, the ratio of cases to controls in the datasets (0.055 in UKB and 0.026 in ICE) are consistent with reported prevalence estimates that use POPQ stage II–IV as a criterion for diagnosis or with symptom-based prevalence estimates[8–11]. The 2-fold difference in the ratio of cases to controls of POP in UKB data compared to the data from Iceland is explained by the difference in age-distributions (Supplementary Data 1). To further compare the POP phenotype between Iceland and UKB, we compared the percentage of cases within each of the nine subgroups of ICD10 code N81. In both datasets the majority of cases are diagnosed with cystocele or rectocele and fewer with uterovaginal prolapse (Supplementary Data 28).

The leiomyoma[38], benign prostatic hyperplasia[61], prostate cancer[61], endometriosis[37], diverticular disease[88], and lung function[70] datasets have been described previously. We obtained blood pressure measurements from Landspitali – The National University Hospital of Iceland, the Primary Health Care Clinics of the Reykjavik area and at recruitment for deCODE studies. Data on gestational duration (GD) (Iceland only) for the first available pregnancy was obtained from the National Birth Registry, excluding multiple pregnancies, stillbirths and pre and post term births (GD ≤259 days and GD ≥301 days) and adjusting for age of mother, sex, and year of birth of the child. Information on number of children was extracted from the deCODE geneology database, adjusting for year of birth and county. Information on BMI was corrected for year of birth, age and county, conditional on age >18. BMI values originate from measured and self-reported data and are mean values for multiple measures within individuals. Waist circumference measurement was adjusted for age, gender, and BMI, conditional on age > 18. The BMD values at head (DEXA, Hologic QDR4500A) were age, height and body mass index corrected (Iceland only). Quantitative traits were rank-based inverse normal transformed to a standard normal distribution separately for each gender. Other datasets from Iceland originate from deCODE genetics phenotype database which contains extensive medical information on various diseases and other traits. Cases with hernias were identified by hospital-based ICD10 diagnosis codes: K40 (inguinal hernia), K41 (femoral hernia), K43 (ventral hernia) in both datasets; Iceland and UKB and the same applies for diverticular disease (K57), stress incontinence (N393), carpal tunnel syndrome (G560), Barrett's oesophagus and esophageal adenocarcinoma combined (K227 and C15, data in UKB only), and actinic keratosis (L57). For sex specific phenotypes in Iceland, only controls for the relevant sex were included in the analysis.

The study was approved by the Icelandic National Bioethics Committee (bioethics consent number VSN 18-067) in agreement with conditions issued by the Data Protection Authority of Iceland. Written informed consent was obtained from all genotyped subjects. Personal identities relating to participants' data and biological samples (i.e. blood samples, buccal samples, or adipose tissue samples) were encrypted by a third-party system (Identity Protection System), approved and monitored by the Data Protection Authority[89]. DNA was extracted from blood and buccal samples and RNA from blood and adipose tissue samples.

**Genotyping and imputation.** Details of genotyping and imputation methods in the Icelandic part of the study have been described[90]. In brief, the whole genomes of 15,220 Icelanders, participating in various disease projects at deCODE genetics, were sequenced to a mean depth of at least 10× (median 32×) using Illumina technology. Genotypes of SNPs and indels were called using joint calling with the Genome Analysis Toolkit HaplotypeCaller (GATK version 3.4.07)[91]. As all of the sequenced individuals had also been chip-typed and long-range phased, informa-tion about haplotype sharing was used to improve genotype calls. In total, 151,677 Icelanders have been genotyped with various Illumina SNP chips, long-range phased and imputed based on the sequenced data set[92]. Using genealogic infor-mation, genotype probabilities for 282,894 untyped relatives of the genotyped individuals were calculated to further increase the sample size for association analysis and to increase the power to detect associations.

The informativeness of genotype imputation was estimated by the ratio of the variance of imputed expected allele counts and the variance of the actual allele counts:

$$\frac{\text{Var}(E(\theta|\text{chip data}))}{\text{Var}(\theta)}$$

where $\theta$ is the allele count. Here, $\text{Var}(E(\theta|\text{chip data}))$ is estimated by the observed variance in the imputed expected counts and $\text{Var}(\theta)$ was estimated by $p\,(1-p)$, where $p$ is the allele frequency. Variants were annotated using Ensembl release 80 and Variant Effect Predictor (VEP) version 2.8[93].

Genotyping of UKB samples was performed using a custom-made Affymetrix chip, UK BiLEVE Axiom[94] in the first 50,000 participants and with the Affymetrix UK Biobank Axiom array[95] in the remaining participants with 95% of the signals on both chips. Imputation was performed by Wellcome Trust Centre for Human Genetics using a combination of the Haplotype Reference Consortium (HRC), 1000 Genomes phase 3 and the UK10K haplotype resources[86]. This yields a total of 92.6 million imputed variants, however, we used only markers that were imputed on the basis of the HRC panel owing to problems in the UK10K + 1000 Genomes panel imputation, resulting in 37.1 million imputed variants used in the current study.

**GWAS and meta-analysis.** We used logistic regression assuming an additive model in the case-control analysis to test for association between variants and disease, treating disease status as the response and expected genotype counts from imputation as covariates, and using likelihood ratio test to compute *P*-values. For the Icelandic data, the model also includes as nuisance variables available indi-vidual characteristics that correlate with phenotype status. In Iceland, these covar-iates are: county of birth, current age or age at death (first and second order terms included), availability of blood sample for the individual and an indicator function for the overlap of the lifetime of the individual with the time span of phenotype collection. The association analysis for both the Icelandic and UKB datasets was done using software developed at deCODE genetics[90]. For the association testing in UKB, 40 principal components were used to adjust for population substructure and age was included as a covariate in the logistic regression model.

To account for inflation in test statistics due to cryptic relatedness and stratification, we applied the method of linkage disequilibrium (LD) score regression[33]. LD scores were downloaded from an LD score database (ftp://atguftp.mgh.harvard.edu/brendan/1k_eur_r2_hm3snps_se_weights.RDS; accessed 23 June 2015). With a set of 1.1 million variants, we regressed the $\chi^2$ statistics from our GWAS scan on the LD scores and used the intercept as a correction factor. *P* values were then adjusted by dividing the corresponding $\chi^2$ values by the correction factors. The estimated correction factor for POP based on LD score regression was 1.12 for the Icelandic and 1.05 for the UK datasets, respectively.

Variants in the UKB imputation dataset were mapped to National Center for Biotechnology Information (NCBI) Genome Reference Consortium Build 38 positions (GRCh38) and matched to the variants in the Icelandic dataset based on allele variation. Only sequence variants from the Haplotype Reference Consortium panel (HRC) were included in the meta-analysis. The results from the two datasets

were combined using a fixed effect model (a Mantel-Haenszel model)[96] in which the datasets were allowed to have different population frequencies for alleles and genotypes but were assumed to have a common OR and weighted with the inverse of the variance. We selected a threshold of 0.8 imputation info and MAF >0.01% for variants available in the Icelandic data set and/or the UKB data set. A total of 37,144,863 variants were used in the analysis. We tested for heterogeneity by comparing the null hypothesis of the effect being the same in both populations to the alternative hypothesis of each population having a different effect using a likelihood ratio test (Cochran's Q) reported as $P_{het}$ (Table 1 for POP and Supplementary Data 29 for other traits).

We used the weighted Bonferroni method to account for all 37,144,863 variants being tested ($P$-value < (0.05 × weight)/37,144,863). Using the weights given in Sveinbjornsson et al.[32], this procedure controls the family-wise error rate at 0.05; $P \leq 2.2 \times 10^{-7}$ for high-impact variants (including stop-gained, frameshift, splice acceptor or donor and initiator codon variants, $N = 9{,}658$), $P \leq 4.4 \times 10^{-8}$ for missense, splice-region variants and in-frame-indels ($N = 180{,}803$), $P \leq 4.0 \times 10^{-9}$ for low-impact variants (including synonymous, 3′ and 5′ UTR, and upstream and downstream variants, $N = 2{,}653{,}622$), $P \leq 6.7 \times 10^{-10}$ for intron and intergenic variants ($N = 34{,}300{,}780$) (Supplementary Data 2). All $P$-values are two-sided.

**Conditional analysis.** We applied approximate conditional analyses, implemented in the GCTA software[97] to the meta-analysis summary statistics to look for additional association signals at each of the genome-wide significant loci. LD between variants was estimated using a set of 8700 whole-genome sequenced Icelandic individuals. The analysis was restricted to variants present in both the Icelandic and UKB datasets and within 1 Mb from the index variants. We tested 7 loci and about 100,000 variants in the conditional analysis and report one variant (rs1430191) with conditional $P$ value < $5 \times 10^{-7}$. The results from GCTA were verified by conditional analysis using genotype data in the Icelandic and UK datasets separately and results presented in Table 1 are obtained by meta-analyzing those results.

**Co-localization analysis.** The variants explored as potentially representing the same signal as the POP variants were found by two means. First, we looked up correlates of the POP variants in the GWAS-catalog ($r^2 > 0.5$) (Supplementary Data 10) and extracted those with $0.5 < r^2 < 0.9$ for the co-localization analysis because of the limited potential to distinguish between variants in LD of $r^2 > 0.9$. Second, for each of the traits associating with the POP variants in our data we extracted the strongest associating variant ($r^2 < 0.9$) for the adjusted (conditional) analysis. For the tests performed in the co-localization analysis, we use previously reported variants as index variants for the secondary traits when available for a conditional analysis, given that the reported analyses are based on a similar sized or a larger sample than our combined data from Iceland and UKB. Otherwise we use the variant most strongly associating with the trait in our data. The results from the conditional analyses are consistent with a single signal representation if two conditions are met: First, the $P$-value of the index variant for the secondary trait is ≥0.05 after adjusting for the POP variant at the locus and second, the POP variant at the locus associates with the secondary trait at a $P$-value < $6.3 \times 10^{-5}$. The latter condition holds for traits identified through the phenomescan (PheWAS) but a $P$-value < 0.05 is required if a correlate of the POP variant is previously reported to associate with the trait in a considerably larger sample than in our data (Supplementary Data 12).

**RNA sequencing analysis.** Generation of poly(A)+ complementary DNA sequencing libraries, RNA sequencing, and data processing were carried out as described previously[98,99]. Two tissue types were available for this analysis: whole blood ($N = 13{,}162$) and adipose tissue ($N = 749$). We used generalized linear regression to test for association between sequence variants and rank-transformed gene expression estimates.

**Polygenic scores, heritability, and functional annotation.** We calculated two sets of polygenic risk scores (PRSs_POP) both using Icelandic and UK data essentially as previously described[100]. Briefly, the PRSs were calculated using genotypes for about 630,000 autosomal markers included on the Illumina SNP chips to avoid uncertainty due to imputation quality. We estimated linkage disequilibrium (LD) between markers using 14,938 phased Icelandic samples and used this LD information to calculate adjusted effect estimates using LDpred[100,101]. To avoid overfitting due to population substructure, the effect estimates calculated using the Icelandic data were used as weights when generating the weighted PRS (PRS_POPICE) for testing in the UK, and the effect estimates generated from the UK data were used to derive the weighted PRS (PRS_POPUKB) for testing in Iceland. We created several PRSs assuming different fractions of causal variants (the P parameter in Ldpred), and selected the best PRS based on prediction of POP in the Icelandic and UK datasets (1% causal variants). The most predictive PRS_POPICE was then used to calculate the correlation with selected phenotypes in the UKB data, and the most predictive PRS_POPUKB was tested for correlation with the selected phenotypes in Iceland. The correlation between the PRS and phenotypes was calculated using logistic regression in R (v3.5) (http://www.R-project.org) adjusting for year of birth and principle components by including them as covariates in the analysis. We

summarize the correlation of the two sets of PRS scores calculated with the Icelandic and UKB phenotypes as weighted average of the effect estimates from both analyses.

Using precomputed LD scores for about 1.1 million variants found in European populations (downloaded from: https://data.broadinstitute.org/alkesgroup/LDSCORE/eur_w_ld_chr.tar.bz2), we estimated SNP heritability with LD score regression[33].

Chromatin interaction map data were derived from Hi-C sequencing for selected cell- and tissue types[79]. The data were downloaded from Omnibus, accession number GSE87112, in pre-processed format (Fit-Hi-C algorithm) representing false-discovery rates (FDR) for contact regions at 40 kb resolution. To define statistically significant contacts we used a threshold value of FDR <10$^{-6}$ in relevant cell-types and tissue (IMR90 fibroblasts, mesenchymal stem cells, and muscle tissue i.e. left ventricle, right ventricle, and psoas muscle)[79]. DNA contact regions containing the lead variant, or those containing variants in LD ($r^2 > 0.8$) to the lead variant, were then identified to find interacting target genes. Using JEME (joint effect of multiple enhancers)[80], we similarly looked for enhancer elements containing the lead variant (and those in LD) to find target genes using similar cell- and tissue types (IMR90 fibroblasts, mesenchymal stem cells, and muscle tissue i.e. colon, stomach, duodenum, and male skeletal muscle). The intersection of genes identified by Hi-C and JEME are then regarded as strong candidate gene targets.

**Reporting summary.** Further information on research design is available in the Nature Research Reporting Summary linked to this article.

## Data availability
The Icelandic population WGS data has been deposited at the European Variant Archive under accession code PRJEB15197. We declare that the data supporting the findings of this study are available within the article, its Supplementary Data files and upon reasonable request. A reporting summary for this Article is available as a Supplementary Information file. The UK Biobank data can be obtained upon application (ukbiobank.ac.uk). The genome-wide association scan summary data will be made available at http://www.decode.com/summarydata.

## Code availability
We use the following publicly available software in conjunction with the above-described algorithms in the sequencing processing pipeline (whole-genome sequencing, association testing, RNA-sequencing mapping and analysis): BWA 0.7.10 mem, https://github.com/lh3/bwa; GenomeAnalysisTKLite 2.3.9, https://github.com/broadgsa/gatk/; Picard tools 1.117, https://broadinstitute.github.io/picard/; SAMtools 1.3, http://samtools.github.io/; Bedtools v2.25.0-76-g5e7c696z, https://github.com/arq5x/bedtools2/; Variant Effect Predictor, https://github.com/Ensembl/ensembl-vep; BOLT-LMM, https://data.broadinstitute.org/alkesgroup/BOLT-LMM/downloads/; LDSC (LD Score), https://github.com/bulik/ldsc.

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

## Acknowledgements

This research has been conducted using the UK Biobank Resource under Application Number 24898 and 24711. We thank the individuals who participated in the study and whose contribution made this work possible. Folkert Asselbergs is supported by UCL Hospitals NIHR Biomedical Research Centre.

## Author contributions

T.O, T.R., G.T., D.F.G., P.S., U.T., and K.S. designed the study and interpreted the results. H.M., K.O., O.I., V.G., and K.J. carried out the subject ascertainment, recruitment, and collection of clinical data. U.S., V.S., J.G., I.J., H.H., V.T., F.W.A., A.O., A.S., and S.L. collected, processed, and analyzed the genotype and phenotype data. T.O., G.T., L.J., J.K. S., A.O., O.A.S., R.P.K., M.L.F, S.L., B.V.H., G.H.H, H.P.E., P.M., P.S., and D.F.G. performed the statistical and bioinformatics analyses. T.O., T.R., D.F.G., G.T., P.S., U.T., and K.S. drafted the manuscript. All authors contributed to the final version of the paper.

## Competing interests

The authors that are affiliated with deCODE are employees of deCODE genetics/Amgen. The remaining authors declare no competing interests.
