## [Peer Review File · Communications Biology]

Reviewers' comments:

Reviewer #1 (Remarks to the Author):

Olafsdottir et al. in their manuscript entitled "Sequence variants associated with pelvic organ prolapse" performed a GWAS of POP using data from Iceland deCODE and UK Biobank for a total of 15,010 cases, the largest study of POP to date. They found 8 variants at seven loci to associate with POP. Investigation of traits with signals in LD with these 8 variants identified 11 other phenotypes, that perhaps share similar pathogenesis, including estrogen exposure and aberrant connective tissue function. This is an interesting paper that provides novel insight into the etiology of POP. However, this study did not replicate any previous gene/linkage/GWAS/meta-analysis finding and did not provide internal replication of their findings.

- 1) The authors list risk factors for POP in the Introduction. Another risk factor that is not mentioned and has been shown to have a strong effect is family history.
- 2) The authors chose to use ICD9 and ICD10 codes to indicate a diagnosis of POP. These codes are notoriously poor at selecting cases – there are many false positives. Procedure codes (e.g., CPT codes) in addition to the ICD9/10 diagnosis codes may perform better in selecting cases. The study would benefit from a sensitivity analysis using surgically treated POP cases to confirm results, particularly for comparison with other previous studies that investigated surgically treated POP.
- 3) The authors investigated ~800 traits in their combined database analysis from Iceland and UK Biobank. It appears that the authors assume that the results are interchangeable; although this likely cannot be assumed as there are differences in ethnicity and hence allele frequency differences. What mitigating factors have the authors employed to control for differences in allele frequencies and ethnicity in their comorbidity analyses?
- 4) The authors investigated possible pleiotropy with a variety of associated traits. The authors should report previous literature findings showing association of these traits within POP cases and their family members to provide confirmation that these traits do share possible variants with POP. For traits such as prostate cancer and BPH, does the literature show that male family members are at increased risk for these diseases?
- 5) The authors stated that controls were recruited through different genetic research projects. Were the controls free from POP?
- 6) The QQ plot for the UK Biobank data appears to greatly deviate from the expected null distribution. What is the genomic inflation factor for this analysis? There may be some cryptic relatedness, undetected population stratification, or systematic bias or other resulting in inflated values. This should be carefully evaluated.
- 7) The authors stated that they extracted correlates with the POP variants in the range of $0.8 < r^2 < 0.9$. Stress incontinence is included in this list and its r^2 value is 0.640.

Reviewer #2 (Remarks to the Author):

This study reports the first evidence of genome-wide significant associations with pelvic organ prolapse (POP). POP is a common serious condition that can result in urinary or bowel symptoms, sexual dysfunction and can affect the quality of life for many women. It has a high prevalence and the heritability is estimated at 43% in twin studies. Analysis of 5,010 cases with hospital-based diagnosis code and 340,734 female controls from Iceland and the UK Biobank identified eight genomic variants at seven loci significantly associated with POP. The loci implicated in POP overlap with other reproductive disorders and with genes involved with connective tissue homeostasis. The authors conclude the results suggest a role of estrogen exposure and connective tissue metabolism in the

etiology of POP.

The discovery of genetic risk factors for POP is an important advance for the field. What is the estimate for SNP heritability for the for POP and how does this compare with the estimated twin heritability?

The authors report interesting overlaps for some regions with related traits and relevant biology for regional candidate genes. However, none of the sentinel SNPs was highly correlated with the top cis-eQTL for neighbouring genes in any of the available tissues using mRNA expression data from the GTEx database and RNA-sequence data on Icelandic samples from blood. There are eQTLs associated with some sentinel SNPs reported the large blood eQTL study (eQTLGen, <https://www.eqtlgen.org/index.html>). These include strong eQTLs for the long non-coding RNA LINC00339 and for the adjacent gene CDC42. There is also evidence for chromatin looping between the region of the lead SNP rs3820282 and the promoter of LINC00339 with a secondary signal for CDC42 (Powell et al. 2016, Human Molecular Genetics: 25, 5046). While WNT4 is an attractive candidate, this may not be the target gene for the signal detected.

Are there other relevant eQTL datasets that could provide evidence to support the role of potential connective tissue candidate genes in POP?

There is repetition between the results and discussion and it may help the structure of the paper if some of the material presented in the results is moved to the discussion.

Reviewer #3 (Remarks to the Author):

The authors performed a genome-wide association study (GWAS) of pelvic organ prolapse among individuals of European ancestry by combining two large datasets - Iceland (3409 cases & 131,444 controls) and UK Biobank (11,601 cases & 209,288 controls). They identified seven loci at genome-wide significance, including one loci with two independent index signals, for a total of eight sequence variants. These eight significant variants were assessed for co-localization with additional traits to further strengthen the biological plausibility of the findings. A few of the novel loci are suspected to be involved with connective tissue homeostasis or estrogen sensitive.

Overall, the current manuscript is strong and presents several novel findings that may have a biologic impact on the field. The authors thoroughly present their findings including their supplementary analyses in both tables and figures. The manuscript is well-organized and clear to understand.

However, a few concerns exist as outlined below.

Major:

1) The study-specific marginal analyses were limited due to omitting potential confounders, impacting interpretations of the overall findings. The literature, including the this manuscript, cites several known risk factors for POP including obesity, age, number of children and number of vaginal births. The current analyses excludes these in their modeling except for age. In supplementary table 1, the

BMI average is presented for each study by case/control status on a subset of the data, although substantial enough to include in a model. Interpreting the genetic results without known POP risk factors is difficult and should be included as well as discussed as a potential weakness.

2) In table 1, several of the major findings have heterogeneous effects by study. A comment should be included on how this impacts the overall interpretation.

Minor:

1) The abstract should define 'common' and include a statement defining statistical significance.

Sequence variants associating with pelvic organ prolapse

Response to Reviewers' Comments

Reviewer #1

Olafsdottir et al. in their manuscript entitled "Sequence variants associated with pelvic organ prolapse" performed a GWAS of POP using data from Iceland deCODE and UK Biobank for a total of 15,010 cases, the largest study of POP to date. They found 8 variants at seven loci to associate with POP. Investigation of traits with signals in LD with these 8 variants identified 11 other phenotypes, that perhaps share similar pathogenesis, including estrogen exposure and aberrant connective tissue function. This is an interesting paper that provides novel insight into the etiology of POP. However, this study did not replicate any previous gene/linkage/GWAS/meta-analysis finding and did not provide internal replication of their findings.

1.1) The authors list risk factors for POP in the Introduction. Another risk factor that is not mentioned and has been shown to have a strong effect is family history.

Response: We now list family history as a one of the reported risk factors for POP, citing Lince, S. L. et al. (*Int Urogynecol J*, **23** (2012)) on p. 4 as follows:

"For all stages of POP, Hispanic ethnicity⁶, lack of pelvic floor muscle strength¹⁸ and family history¹⁹ are also reported as risk factors, and previous hysterectomy is associated with severe POP²⁰."

1.2) The authors chose to use ICD9 and ICD10 codes to indicate a diagnosis of POP. These codes are notoriously poor at selecting cases – there are many false positives. Procedure codes (e.g., CPT codes) in addition to the ICD9/10 diagnosis codes may perform better in selecting cases. The study would benefit from a sensitivity analysis using surgically treated POP cases to confirm results, particularly for comparison with other previous studies that investigated surgically treated POP.

Response: We used the ICD-codes to indicate a diagnosis of POP because of very limited information on POP procedure codes in the Iceland data, with procedure-diagnosed POP cases corresponding to 13% of POP cases with ICD-codes (437/3,409). However, as we have information on procedure codes for 61% of the ICD-coded POP cases (7,121/11,601) in the UKB we can check the robustness of our results for the UKB data as suggested by the reviewer. The classification of surgical procedures used in the UK is OPCS-4 (Office of Population Censuses and Surveys classification of surgical operations and procedures), with procedure codes P22, P23 and P24 being surgical treatments for POP (**Supplementary Table 7**). We extracted cases having

both those procedure codes and the ICD10 code N81 from data on inpatient episodes in the UK Biobank (HESIN-table, Category 2000, resource 138483). We then tested the eight POP variants for association with POP defined by procedure codes in the UKB. We did the same in Iceland for completeness, albeit with limited statistical power. As can be seen in **Supplementary Table 8**, the ORs for the ICD10- and procedure-diagnosed POP cases in UKB (see column named “ICD10 with procedure”) all have same direction of an effect and tend to have larger effects than those based only on ICD codes. The greater effect size, may be reflecting greater severity of POP among procedure-diagnosed POP cases or a greater number of falsely diagnosed POP cases among the ICD based cases. Based on this analysis, using the available data on surgical codes from UKB (covering the majority (61%) of POP cases diagnosed with ICD10 code), we conclude that our results in **Table 1** are robust and not substantially different from those using just surgically treated POP cases.

We now report the results from this analysis in the text on p. 6-7 as follows:

“We assessed the robustness of the POP association results for the eight lead variants in a group of more stringently diagnosed POP cases that are defined by procedure codes specific to the treatment of POP in addition to the ICD10 code N81 (**Supplementary Table 7**). The statistical power for this analysis comes mainly from the UKB data because information on POP related surgeries was available for 61% of the ICD-coded POP cases in UK, but only 13% of the Icelandic POP cases. The effect sizes tended to be greater for the association using cases that had undergone surgery compared to those diagnosed only based on ICD code (**Supplementary Table 8**). This most likely reflects increased severity of POP among those that had undergone surgery or a greater number of falsely diagnosed POP cases among the ICD based cases. We conclude that our results based on ICD codes alone are robust as the effects are in the same direction and the effect sizes are not substantially different from those using surgically treated POP cases.”

1.3) The authors investigated ~800 traits in their combined database analysis from Iceland and UK Biobank. It appears that the authors assume that the results are interchangeable; although this likely cannot be assumed as there are differences in ethnicity and hence allele frequency differences. What mitigating factors have the authors employed to control for differences in allele frequencies and ethnicity in their comorbidity analyses?

Response: To reduce the risk of confounding due to differences in ancestral background in the data from UKB, the cases and controls from UKB used in the study are self-reported white British individuals with similar genetic ancestry based on principal component analysis (see details in Bycroft, C. et al, 2017) (see **Methods**, p. 17). We performed an association test in each population separately and then combined the results using meta-analysis based on a fixed effects model. This model only assumes that the effects are the same in the two populations – there is no assumption of identical allele frequencies between the two populations. To validate this assumption, we tested for heterogeneity in effect estimates between the populations using

a likelihood ratio test (Cochran's Q) and now report these results for the eight POP variants by traits in **Supplementary Table 29** (P_{het}). Adjusting for number of tests performed within each trait ($0.05/8=0.0063$), the P_{het} is below the Bonferroni-adjusted threshold for rs72624976 ($7q32.1$, $P_{\text{het}}=0.0029$) and rs9306894 ($2p24.1$, $P_{\text{het}}=0.0059$) when tested against FEV1/FVC. Also at the margin are variants rs3820282 ($1p36.12$, carpal tunnel syndrome, $P_{\text{het}}=0.0059$) and rs3791675 ($2p16.1$, diverticular disease, $P_{\text{het}}=0.0051$).

We now refer to **Supplementary Table 29** in the **Methods** section on p. 21-22 as follows:

"We tested for heterogeneity by comparing the null hypothesis of the effect being the same in both populations to the alternative hypothesis of each population having a different effect using a likelihood ratio test (Cochran's Q) reported as P_{het} (**Table 1** for POP and **Supplementary Table 29** for other traits)."

1.4) The authors investigated possible pleiotropy with a variety of associated traits. The authors should report previous literature findings showing association of these traits within POP cases and their family members to provide confirmation that these traits do share possible variants with POP. For traits such as prostate cancer and BPH, does the literature show that male family members are at increased risk for these diseases?

Response: We are not aware of previous literature findings showing associations of these traits within POP cases and their family members. Such findings might be helpful as a way to confirm that these traits do share variants with POP. However, since all of the POP variants are common variants with ORs in the range of 0.8-1.13, those variants cannot generate much familial risk. We have added a sentence on this for clarity in the text on p. 12:

"We are not aware of previous literature findings showing associations of these phenotypes within POP cases and their family members."

1.5) The authors stated that controls were recruited through different genetic research projects. Were the controls free from POP?

Response: Controls have not been diagnosed by hospital-based ICD10 code N81 or ICD9 code 618 in Iceland and ICD10 N81 in UKB. Given the prevalence of symptom-based or POPQ stage II-IV criterion for diagnosis being in the range of 3-26%, and considering the proportion of cases in our data (5.5% in UKB and 2.6% in Iceland) it is clear that a fraction of the controls in both datasets will have undiagnosed POP cases, diagnosed POP-cases that we do not have information on or individuals that have not yet developed the condition. This under-diagnosis of the controls weakens the power of the GWAS and will lead to underestimation of effect sizes, but will not lead to false-positive results.

1.6) The QQ plot for the UK Biobank data appears to greatly deviate from the expected null distribution. What is the genomic inflation factor for this analysis? There may be some cryptic relatedness, undetected population stratification, or systematic bias or other resulting in inflated values. This should be carefully evaluated.

Response: The significant (true) associations with POP drive a deviation from the null distribution of the test-statistic. This holds after adjustment of the test statistics for inflation due to cryptic relatedness and stratification. We applied the method of linkage disequilibrium (LD) score regression (see **Methods** for further details and Bulik-Sullivan, B. K. et al. (*Nat Genet*, **47** (2015))) and the estimated correction factor for POP was 1.05 for the UK dataset. The chi-square test statistics were thus divided by 1.05 to correct for inflation in the significance due to cryptic relatedness and stratification for variants found in the UK dataset (See **Methods**, p. 21). Furthermore, for the association testing in UKB, 40 principal components were used to adjust for population substructure. The large difference observed in the deviation from the expected null distribution between Iceland and the UK populations in the QQ plots can be explained by the difference in statistical power between the datasets from Iceland and UKB (3,409 POP cases in Iceland and 11,601 POP cases in UKB). To further elucidate the reason for the large deviation of *P*-values from the expected null distribution seen in the QQ-plot of *P*-values for variants in UKB, we have re-generated the QQ-plot after excluding the POP variants and all correlates ($r^2 > 0.05$) within a 2MB window (see the following **Figure**). As can be seen in the figure below there is little deviation of the distribution of the test statistic from the null distribution. We have added this figure to **Supplementary Figure 1** and changed the figure legend accordingly:

Changes are underlined:

“Supplementary Figure 1 A quantile-quantile plot (QQ-plot) of the P-values (chi-square statistics corrected for relatedness and stratification using correction factor estimated from LD score regression (see Methods)) for variants from the GWAS of pelvic organ prolapse in a) Iceland, b) UK Biobank and c) UK Biobank excluding the eight lead POP variants and their correlates of $r^2 > 0.05$ within a 2MB window. The reason for the large difference observed in the deviation of the P-values from the expected null distribution in a) and b) is the difference in statistical power between the datasets from Iceland and UKB (3,409 POP cases in Iceland and 11,601 POP cases in UKB). As shown in c), once the POP variants and their correlates are excluded from the QQ plot there is little deviation of the distribution of the test statistic from the null distribution. Sequence variants with imputation information > 0.8 and minor allele frequency $> 0.01\%$ are plotted in the figure. The red diagonal line represents no departure of the empirical (observed) distribution from the expected distribution of the chi-square statistics.”

We also want to note that we made an additional revision to the figure legend as previously it was stated that we used genomic control to correct for relatedness but for clarity we now

explain that we used LD score regression to correct for relatedness and stratification as described in the **Methods** section.

Figure: Shown are variants in UK Biobank excluding the eight lead POP variants and their correlates of $r^2 > 0.05$.

1.7) The authors stated that they extracted correlates with the POP variants in the range of $0.8 < r^2 < 0.9$. Stress incontinence is included in this list and its r^2 value is 0.640.

Response: How variants correlated with our POP variants were identified when scanning for association with other traits in the GWAS-catalog was confusing in comparison with how variants were identified through the scanning for traits associating with the lead POP variants in our own data. For clarity, we now use the threshold of $r^2 > 0.5$ for correlates of the POP variants in the GWAS-catalog instead of the previous threshold of $r^2 > 0.8$ to reconcile the two ways of identifying variants subsequently used in the exploration of single signal representations across traits. For the co-localization analysis, we set the upper limit of $r^2 < 0.9$ because of the limited potential to distinguish between variants in LD of $r^2 > 0.9$. By changing the r^2 filter from 0.8 to 0.5 when looking up variants in the GWAS-catalog, seven additional correlates of the POP variants were detected of which five associate with four traits not previously listed in **Supplementary Table 10**. Those are paediatric bone density of the skull, heel bone mineral density, paediatric bone mineral density (total body less head) and esophageal adenocarcinoma (combined with Barrett's esophagus). Co-localization analysis of signals for POP and esophageal adenocarcinoma combined with Barrett's esophagus, – the trait for which we have data – suggests that the same signal at the 2p24.1 locus associates with these traits. We have now

added this result to **Table 2** and made the corresponding revisions to **Supplementary Tables 9 and 12**.

Furthermore, we now only include in **Supplementary table 10** variants from the GWAS-catalog that correlate with the POP lead variants ($r^2 > 0.5$) if reported to associate with traits with P -value $< 1 \times 10^{-8}$ to reduce the possibility of including false positives in our data analyses. We have made the corresponding changes to the main text on p. 9, 10 and 11 which entails removing discussion of the following traits that associate with a P -value $\geq 1 \times 10^{-8}$ with correlates of our POP variants: Epithelial ovarian cancer ($P = 2 \times 10^{-8}$), joint hypermobility ($P = 1.4 \times 10^{-7}$) and chin dimples ($P = 1 \times 10^{-8}$). We have also revised the table notes in **Supplementary Table 10** according to the change in r^2 filter for the GWAS-catalog look-up and the applied P -value filter of 1×10^{-8} .

Related to the reviewer's comment the explanation of the co-localization analysis needed some clarification. We have now revised the corresponding text in the **Methods** section on p. 23 as follows:

“The variants explored as potentially representing the same signal as the POP variants were found by two means. First, we looked up correlates of the POP variants in the GWAS-catalog ($r^2 > 0.5$) (**Supplementary Table 10**) and extracted those with $0.5 < r^2 < 0.9$ for the co-localization analysis because of the limited potential to distinguish between variants in LD of $r^2 > 0.9$. Second, for each of the traits associating with the POP variants in our data we extracted the strongest associating variant ($r^2 < 0.9$) for the adjusted (conditional) analysis. For the tests performed in the co-localization analysis, we use previously reported variants as index variants for the secondary traits when available for a conditional analysis, given that the reported analyses are based on a similar sized or a larger sample than our combined data from Iceland and UKB. Otherwise we use the variants most strongly associating with the trait in our data. The results from the conditional analyses are consistent with a single signal representation if two conditions are met: First, the P -value of the index variant for the secondary trait is ≥ 0.05 after adjusting for the POP variant at the locus and second, the POP variant at the locus associates with the secondary trait at a P -value $< 6.3 \times 10^{-5}$. The latter condition holds for traits identified through the phenomescan (PheWAS) but a P -value < 0.05 is required if a correlate of the POP variant is previously reported to associate with the trait in a considerably larger sample than in our data (**Supplementary table 12**). ”

Reviewer #2 (Remarks to the Author):

This study reports the first evidence of genome-wide significant associations with pelvic organ prolapse (POP). POP is a common serious condition that can result in urinary or bowel symptoms, sexual dysfunction and can affect the quality of life for many women. It has a high prevalence and the heritability is estimated at 43% in twin studies. Analysis of 5,010 cases with hospital-based diagnosis code and 340,734 female controls from Iceland and the UK Biobank identified eight genomic variants at seven loci significantly associated with POP. The loci implicated in POP overlap with other reproductive disorders and with genes involved with connective tissue homeostasis. The authors conclude the results suggest a role of estrogen exposure and connective tissue metabolism in the etiology of POP.

2.1) The discovery of genetic risk factors for POP is an important advance for the field. What is the estimate for SNP heritability for the for POP and how does this compare with the estimated twin heritability?

Response: Compared to the estimated twin heritability of 43%, the estimated SNP heritability for POP is 12.4%. We have added the following to the text on p. 7:

“We used LD score regression to estimate the SNP heritability of POP³³. Using LD scores for about 1.1 million variants found in European populations we estimated SNP heritability in the meta-analysis to be 12.4% (95% CI 9.9-14.8%).”

The methods section now includes the following on p. 24.

“Using precomputed LD scores for about 1.1 million variants found in European populations (downloaded from: https://data.broadinstitute.org/alkesgroup/LDSCORE/eur_w_ld_chr.tar.bz2), we estimated SNP heritability with LD score regression³³.”

2.2) The authors report interesting overlaps for some regions with related traits and relevant biology for regional candidate genes. However, none of the sentinel SNPs was highly correlated with the top cis-eQTL for neighbouring genes in any of the available tissues using mRNA expression data from the GTEx database and RNA-sequence data on Icelandic samples from blood. There are eQTLs associated with some sentinel SNPs reported the large blood eQTL study (eQTLGen, <https://www.eqtlgen.org/index.html>). These include strong eQTLs for the long non-coding RNA LINC00339 and for the adjacent gene CDC42. There is also evidence for chromatin looping between the region of the lead SNP rs3820282 and the promoter of LINC00339 with a secondary signal for CDC42 (Powell et al. 2016, Human Molecular Genetics: 25, 5046). While WNT4 is an attractive candidate, this may not be the target gene for the signal detected.

Are there other relevant eQTL datasets that could provide evidence to support the role of

potential connective tissue candidate genes in POP?

Response: In our blood RNA sequencing data (N=13,162) the top eQTL for *CDC42* is rs2501299 and in the eQTLGEN database (N=31,567 from 36 cohorts) the top eQTL is rs2473290. Since neither of these variants associate with POP in the combined data set from Iceland and UK after adjusting for the lead POP variant rs3820282 (see **Table** below), it is difficult to conclude that rs3820282 exerts its effect on POP through expression of *CDC42* gene.

SNP Pos Hg38	MAF (%)	POP				CDC42 expression				
		Unadjusted		Adjusted		Unadjusted		Adjusted		
		P	OR	P	OR	P	SD	P	SD	Covariate
rs3820282 1:22141722	19	3.3E-21	0.85	7.0E-21	0.84	4.8E-108	0.36	6.5E-26	0.17	rs2501299
rs2501299 1:22019154	36	0.07	1.02	0.16	1.02	<1E-306	0.50	1.1E-257	0.45	rs3820282
rs3820282 1:22141722	19	2.3E-21	0.85	1.9E-12	0.84	4.8E-108	0.36	2.1E-11	0.14	rs2473290
rs2473290 1:22031964	29	1.0E-10	0.91	0.72	1.01	2.1E-150	0.36	4.8E-54	0.28	rs3820282

Shown are associations of POP variant rs3820282 and the top cis-eQTL in *CDC42* in deCODE whole blood eQTL database (rs2501299) and the top cis-eQTL in *CDC42* reported in the eQTLGen database (rs2473290) with POP in the combined data from Iceland and UKB and with *CDC42* expression in deCODE RNA-sequence data, along with a pairwise conditional analysis for both associations tested. $r^2=0.11$ between rs3820282 and rs2501299. $r^2=0.45$ between rs3820282 and rs2473290. MAF; minor allele frequency (Iceland), SD; standard deviation, OR; odds ratio.

The *LINC00339* gene was missing from Ensembl cDNA database (release 87) and therefore not a part of the transcriptome reference used for gene abundance estimate in the deCODE eQTL analysis. In the eQTLGen data the top cis-eQTL in whole blood for *LINC00339* is rs11801382. Rs11801382 and the POP variant rs3820282 are not correlated ($r^2=0.036$) and rs11801382 does not associate with POP (P -value=0.20). The POP association is thus unlikely to go through effect on *LINC00339*.

Powell, J. E. et al. (*Hum Mol Genet*, **25** (2016)) describe *CDC42* and *LINC00339* as likely target genes for rs3820282 at chromosome 1p36.12 based on co-localization with an eQTL signal found in whole blood samples (N=862), although not the top eQTL. Using chromatin conformation capture (3C), the authors showed direct interaction between a putative regulatory element containing a variant (rs12038474) in LD ($r^2=0.81$) with rs3820282 and

promoter region for *CDC42* in endometrial adenocarcinoma which provides support for this gene as a candidate target gene.

Using the JEME (joint effect of multiple enhancers) resource (Cao, Q. et al, *Nat Genet*, **49** (2017)), which is based on available expression data and histone mark/eRNA profiling indicative of enhancers, we explored whether the POP lead SNP rs3820282, or those in LD, reside in regions that interact with nearby genes. This analysis points to the following genes: *HSPG2*, *CDC42*, *WNT4* and *CELA3A* exploring IMR90 fibroblasts, mesenchymal stem cells, and muscle tissue i.e. colon, stomach, duodenum and male skeletal muscle. We also used another publicly available resource containing multiple tissue and cell-types analysed by Hi-C (Schmitt, A. D. et al, *Cell reports*, **17** (2016)) of which we selected IMR90 fibroblasts, mesenchymal stem cells and muscle tissue, i.e. left ventricle, right ventricle and psoas muscle. That analysis provides support for DNA-DNA contacts between the region containing the rs3820282 and the following genes – also identified using JEME: *WNT4*, *CDC42* and *HSPG2*. In summary, available Hi-C data along with enhancer-gene predictions suggests interactions of rs3820282 not only to *CDC42*, but also to other genes, including *WNT4*.

We have added the following to the text on p. 14.

“A previous report showed chromatin looping between the region containing rs12038474 (LD to rs3820282; $r^2=0.81$) and *CDC42* in endometrial adenocarcinoma⁸¹. Our analysis of available Hi-C data⁸² along with enhancer-gene predictions⁸³ suggests *WNT4*, *HSPG2* and *CDC42* as possible enhancer-gene targets, using relevant cell-types and tissues (see **Methods**). Based on the role of *WNT4* in female sex organs development, *WNT4* can be considered to be a strong candidate gene at the locus.”

We have also added the following to the **Methods** section on p. 25:

“Chromatin interaction map data were derived from Hi-C sequencing for selected cell- and tissue types¹⁰⁵. The data were downloaded from Omnibus, accession number GSE87112, in pre-processed format (Fit-Hi-C algorithm) representing false-discovery rates (FDR) for contact regions at 40 kb resolution. To define statistically significant contacts we used a threshold value of $FDR < 10^{-6}$ in relevant cell-types and tissue (IMR90 fibroblasts, mesenchymal stem cells, and muscle tissue i.e. left ventricle, right ventricle and psoas muscle)¹⁰⁵. DNA contact regions containing the lead variant, or those containing variants in LD ($r^2>0.8$) to the lead variant, were then identified to find interacting target genes. Using JEME (joint effect of multiple enhancers)⁸³, we similarly looked for enhancer elements containing the lead variant (and those in LD) to find target genes using similar cell- and tissue types (IMR90 fibroblasts, mesenchymal stem cells, and muscle tissue i.e. colon, stomach, duodenum and male skeletal muscle). The intersection of genes identified by Hi-C and JEME are then regarded as strong candidate gene targets.”

It is not clear which tissue type/s would be the most relevant to screen for eQTLs having a potential role in POP etiology. For all of the available tissues in the GTEx database, which include ovary, vagina, uterus and fibroblasts, none of the top eQTLs was highly correlated with our POP variants. We are not aware of relevant eQTL datasets other than the three described here; our RNA-sequence eQTL data, the eQTLGen database and the GTEx database.

2.3) There is repetition between the results and discussion and it may help the structure of the paper if some of the material presented in the results is moved to the discussion.

We have moved some of the text regarding the 1p36.12 locus presented in the results to the discussion. After adding the in-text paragraph from the response to comment 2.2 (underlined) to that section, the previously presented text in the results;

“Of the 23 correlated variants at the 1p36.12 locus, rs3820282 has the best functional candidacy. Rs3820282–T resides in a region marked by mono-methylation of histone H3 at lysine residue K4 (H3K4me1), a characteristic of enhancers, and is predicted to alter a conserved binding site for the transcription factors (TFs) ESR1 and ESR2³⁶ supported by ChIP-seq data in breast and bone cell lines³⁷. This suggests that the variant may alter the estrogen-based regulation of *WNT4* or adjacent genes³⁸. The role of *WNT4* in female sex organ development³⁹, makes it a strong candidate gene at the locus.”

is now in the discussion on p. 14 as follows:

“Of the 23 correlated variants at the 1p36.12 locus, rs3820282 has the best functional candidacy. Rs3820282–T resides in a region marked by mono-methylation of histone H3 at lysine residue K4 (H3K4me1), a characteristic of enhancers, and is predicted to alter a conserved binding site for the transcription factors (TFs) ESR1 and ESR2³⁶ supported by ChIP-seq data in breast and bone cell lines³⁷. This suggests that the variant may alter the estrogen-based regulation of *WNT4* or adjacent genes³⁸. A previous report showed chromatin looping between the region containing rs12038474 (LD to rs3820282; $r^2=0.81$) and *CDC42* in endometrial adenocarcinoma⁸¹. Our analysis of available Hi-C data⁸² along with enhancer-gene predictions⁸³ suggests *WNT4*, *HSPG2* and *CDC42* as possible enhancer-gene targets, using relevant cell-types and tissues. Based on the role of *WNT4* in female sex organ development³⁹, *WNT4* can be considered to be a strong candidate gene at the locus.”

Reviewer #3 (Remarks to the Author):

The authors performed a genome-wide association study (GWAS) of pelvic organ prolapse among individuals of European ancestry by combining two large datasets - Iceland (3409 cases & 131,444 controls) and UK Biobank (11,601 cases & 209,288 controls). They identified seven loci at genome-wide significance, including one loci with two independent index signals, for a total of eight sequence variants. These eight significant variants were assessed for co-localization with additional traits to further strengthen the biological plausibility of the findings. A few of the novel loci are suspected to be involved with connective tissue homeostasis or estrogen sensitive.

Overall, the current manuscript is strong and presents several novel findings that may have a biologic impact on the field. The authors thoroughly present their findings including their supplementary analyses in both tables and figures. The manuscript is well-organized and clear to understand.

However, a few concerns exist as outlined below.

Major:

3.1) The study-specific marginal analyses were limited due to omitting potential confounders, impacting interpretations of the overall findings. The literature, including the this manuscript, cites several known risk factors for POP including obesity, age, number of children and number of vaginal births. The current analyses excludes these in their modeling except for age. In supplementary table 1, the BMI average is presented for each study by case/control status on a subset of the data, although substantial enough to include in a model. Interpreting the genetic results without known POP risk factors is difficult and should be included as well as discussed as a potential weakness.

Response: We do not see evidence of a causal relationship at the genetic level between POP and two commonly reported risk factors for POP; BMI and number of children. Firstly, we explored whether sequence variants that are associated with BMI and number of children associate with POP. We tested 152 BMI variants (Locke, A. E. et al., *Nature*, **518**, 197-206 (2015) and Turcot V. et al., *Nat Genet*, **50**, 766-767 (2018)) and 3 fertility variants (Barban, N. et al. *Nat Genet*, **48**, 1462-1472 (2016)), but none associated significantly with POP after accounting for the number of tests performed ($0.05/155=3.2\times 10^{-4}$) (**Supplementary Tables 23 and 24**). Furthermore, we did not observe a correlation between POP- and BMI effect estimates for the BMI variants (**Supplementary Figure 5**). Second, the effect sizes and *P*-values for the eight lead variants were robust to adding BMI and number of children separately as covariates in our POP association models (**Supplementary Table 25**). Third, we plotted effect estimates for POP variants against their effect on BMI (**Supplementary Figure 6**) and number of children (combined data from Iceland and UKB). By conducting the last analysis, we found that one of

the POP variants, rs3820282 at *WNT4* correlates with number of children in the combined data from Iceland and UKB. The POP-protecting allele rs3820282–T associates with fewer number of children (beta=-0.016, P -value= 8.6×10^{-6} (Bonferroni corrected P -value= 6.3×10^{-5}) (**Supplementary Figure 6**). We have now added this finding to **Supplementary Table 9** and revised the text accordingly. Our co-localization analysis did not suggest a single signal representation for POP and number of children (**Supplementary Table 12**). Lastly, using polygenic risk score (PRS) for POP, there is little evidence that genetic variants affect POP through BMI or number of children (**Supplementary Table 26**). We have added a new section to the **Results** on p. 12 that summarizes the abovementioned findings.

“BMI is not associated with POP at the genetic level

We screened for evidence of a genetic relationship of POP and two of the most consistently reported risk factors for POP in addition to age; BMI and number of children. We verified that variants reported to associate with BMI^{78,79} or number of children⁸⁰ do not associate with POP (**Supplementary Tables 23 and 24**) and that adding BMI or number of children separately as covariates in our models did not affect the effect sizes or significance of the eight lead variants when tested for association with POP (**Supplementary Table 25**). Furthermore, no correlation was found between effect estimates of BMI- or number of children sequence variants and their effects on POP and vice versa (**Supplementary Figures 5 and 6**). Using polygenic risk scores for POP, we saw little evidence of association with number of children (P -value= 9.5×10^{-5} , beta=0.015) in Iceland only and none with BMI (**Supplementary Table 26**). For BMI, we found no evidence of causal relationship with POP at the genetic level. With that in mind, and since only one of the POP variants (rs3820282) associates with number of children we do not find strong support for a causal pathway or a common third factor affecting POP and those two traits.”

We have added the following text to the **Methods** section on p. 18 to describe the data on number of children and BMI in Iceland:

“Information on number of children was extracted from deCODE genealogy database, adjusting for year of birth and county. Information on BMI was corrected for year of birth, age and county, conditional on age>18. BMI values originate from measured and self-reported data and are mean values for multiple measures within individuals. Waist circumference measurement was adjusted for age, gender and BMI, conditional on age>18.”

We have also added the following text to the **Methods** section on p. 24 to describe our calculations of polygenic risk scores and phenotype correlation analysis:

“We calculated two sets of polygenic risk scores (PRSs_{POP}) both using Icelandic and UK data essentially as previously described¹⁰³. Briefly, the PRSs were calculated using genotypes for about 630,000 autosomal markers included on the Illumina SNP chips to

avoid uncertainty due to imputation quality. We estimated linkage disequilibrium (LD) between markers using 14,938 phased Icelandic samples and used this LD information to calculate adjusted effect estimates using LDpred^{103,104}. To avoid overfitting due to population substructure, the effect estimates calculated using the Icelandic data were used as weights when generating the weighted PRS (PRS_{POPICE}) for testing in the UK, and the effect estimates generated from the UK data were used to derive the weighted PRS (PRS_{POPUKB}) for testing in Iceland. We created several PRSs assuming different fractions of causal variants (the P parameter in LDpred), and selected the best PRS based on prediction of POP in the Icelandic and UK datasets (1 % causal variants). The most predictive PRS_{POPICE} was then used to calculate the correlation with selected phenotypes in the UKB data, and the most predictive PRS_{POPUKB} was tested for correlation with the selected phenotypes in Iceland. The correlation between the PRS and phenotypes was calculated using logistic regression in R (v3.5) (<http://www.R-project.org>) adjusting for year of birth and principle components by including them as covariates in the analysis. We summarize the correlation of the two sets of PRS scores calculated with the Icelandic and UKB phenotypes as weighted average of the effect estimates from both analyses.”

3.2) In table 1, several of the major findings have heterogeneous effects by study. A comment should be included on how this impacts the overall interpretation.

Response: We tested for heterogeneity in effect estimates between the populations using a likelihood ratio test (Cochran’s Q) reported as P_{het} in **Table 1** for the eight POP variants. Accounting for multiple testing ($P > 0.05/8 = 0.0063$), there is no significant heterogeneity in the effect estimates from the two datasets. For clarity, we now summarize the results from the heterogeneity tests in the following sentence on p. 6.

“All eight variants were nominally significant in both populations and accounting for multiple testing ($P\text{-value} > 0.05/8 = 0.0063$), there is no significant heterogeneity in the effect estimates from the two datasets (**Table 1**).”

Minor:

3.1) The abstract should define 'common' and include a statement defining statistical significance.

Response: We have changed the abstract as suggested.

Remarks from the authors to the editor and the reviewers

We note that in addition to responding to the reviewer's comments, which we have found helpful for improving the paper, we have made three minor corrections to the manuscript:

First, in a previous version of **Supplementary Table 9**, we mistakenly reported results for association of the POP index variants with waist circumference adjusted for birth year and age but we now report results for waist circumference adjusted for birth year, age and BMI in accordance to the trait listed in **Supplementary Table 10** (GWAS-catalog scan) for which a reported variant, rs3791679, correlates with the POP variant rs3791675 at *EFEMP1*. We have made the appropriate correction to **Table 2** (variant rs3791675, reported for waist circumference was substituted with rs3791679 ($r^2=0.95$) reported for BMI-adjusted waist circumference, and pubmedID 28448500 was correspondingly substituted with 25673412). Second, r^2 between a reported variant for gestational duration, rs56318008, and POP variant rs3820282 was wrongly reported as 0.97 but the correct value is 0.88. For that reason we report a conditional analysis of those variants in **Supplementary Table 12**. Third, an error in **Table 1** was corrected as follows: OR in Iceland for variant rs1247943 against POP changes from 1.15 (95% CI 1.06-1.25), P -value= 3.2×10^{-4} to 1.10 (95% CI 1.05-1.17), P -value= 3.3×10^{-4} . A typo in Phet for rs1247943 in **Table 1** was corrected from 0.26 to 0.73.

REVIEWERS' COMMENTS:

Reviewer #1 (Remarks to the Author):

The authors have addressed my concerns.

Reviewer #2 (Remarks to the Author):

I have no further comments